# Knowledge of and preparedness for COVID-19 among Somali healthcare professionals: A cross-sectional study

Jude Alawa[1], Lucas Walz[2]°, Samir Al-Ali[3]°, Nikhil Harle[3], Eleanor Wiles[4], Mohamed Abdullahi Awale[5], Deqo Mohamed[6], Kaveh Khoshnood[2]*

**1** Stanford University School of Medicine, Stanford University, Stanford, California, United States of America, **2** Yale School of Public Health, Yale University, New Haven, Connecticut, United States of America, **3** Yale College, Yale University, New Haven, Connecticut, United States of America, **4** Milken Institute School of Public Health, George Washington University, Washington, District of Columbia, United States of America, **5** SIMAD University Faculty of Medicine and Health Sciences, SIMAD University, Mogadishu, Somalia, **6** Hagarla Institute, Mogadishu, Somalia

☯ These authors contributed equally to this work.
* kaveh.khoshnood@yale.edu

**Data Availability Statement:** All relevant data are within the manuscript and its Supporting Information files.

## Abstract

### Background

Somalia is considered severely underprepared to contain an outbreak of COVID-19, with critical shortages in healthcare personnel and treatment resources. In limited-resource settings such as Somalia, providing healthcare workers with adequate information on COVID-19 is crucial to improve patient outcomes and mitigate the spread of the SARS-CoV-2 virus. This study assessed the knowledge of, preparedness for, and perceptions toward COVID-19 prevention and treatment among Somali healthcare workers.

### Methods

A descriptive, cross-sectional survey was completed by 364 Somali healthcare workers in summer of 2020 utilizing a convenience sampling method.

### Results

Participants' most accessed sources of COVID-19 information were from social media (64.8%), official government and international health organization websites (51.1%,), and traditional media sources such as radio, TV, and newspapers (48.1%). A majority of participants demonstrated strong knowledge of treatment of COVID-19, the severity of COVID-19, and the possible outcomes of COVID-19, but only 5 out of 10 symptoms listed were correctly identified by more than 75% of participants. Although participants indicated seeing a median number of 10 patients per week with COVID-19 related symptoms, access to essential medical resources, such as N95 masks (30.2%), facial protective shields (24.5%), and disposable gowns (21.4%), were limited. Moreover, 31.3% agreed that Somalia was in a good position to contain an emerging outbreak of COVID-19. In addition, 40.4% of participants

**Funding:** Authors JA, LW, SA, AA, DM, and KK received limited funding support from a Global Health Development/EMPHNET EMR Operational Studies and Research Grant (http://emphnet.net/; Ref: ADM/625). The funders had no role in study design, data collection and analysis, decision to publish, or preparation of the manuscript.

**Competing interests:** The authors have declared that no competing interests exist.

agreed that the Somali government's response to the pandemic was sufficient to protect Somali healthcare professionals.

## Conclusion

This study provides evidence for the need to equip Somali healthcare providers with more information, personal protective equipment, and treatment resources such that they can safely and adequately care for COVID-19 patients and contain the spread of the virus. Social media and traditional news outlets may be effective outlets to communicate information regarding COVID-19 and the Somali government's response to frontline healthcare workers.

## Introduction

The 2019 novel coronavirus (COVID-19) causes an infectious respiratory illness that spreads through saliva and nasal droplets [1]. The first cases of COVID-19 surfaced in China at the end of 2019, and the World Health Organization declared the disease a pandemic on March 11th, 2020 [2]. As of April 4th, 2021, there were over 131.1 million confirmed cases of COVID-19 globally [3]. The true extent of COVID-19, however, is difficult to assess in many limited resource countries, where the infrastructure, guidance and resources necessary to control an outbreak are scarce. According to the Global Health Security Index, Somalia ranks 194 out of 195 countries in preparedness for a globally catastrophic biological event [4].

The Somali Ministry of Health announced the nation's first confirmed case of COVID-19 on March 16th, 2020. Though the Ministry has reported approximately 11,500 cases and 530 deaths as of April 2021, it is likely that the true number of cases in Somalia is much higher due to a severe lack of testing capabilities [5,6]. In addition, as a result of more than three decades of violent conflict and natural disasters, the Somali healthcare system has remained extremely weak, poorly resourced, and deeply inequitable [5,6]. The lack of available health resources directly impacts the accessibility and quality of healthcare in Somalia. Along with a significant shortage of healthcare workers, there is a national deficit in essential equipment to treat severe COVID-19 infections, with no ventilators and only two intensive care units available across the entire country [7–10]. Similarly, only half of Somali health centers have consistent access to electricity [10,11]. A 2017 report found that, at only $33 per person, Somalia spends the least on healthcare out of 184 countries examined [12].

In order to control the spread of COVID-19 in limited-resource settings, preventative measures and adequate education are crucial. To address the threat of COVID-19 in Somalia, many organizations have scaled up water, sanitation, and hygiene (WASH) services, increased surveillance, and developed awareness campaigns [6,13–16]. In the same way, it is important that healthcare workers in Somalia are equipped with the knowledge and skills necessary to prevent the spread of COVID-19 and treat existing cases appropriately [17]. Some previous studies have shown that healthcare workers had a lack of knowledge and attitude toward MERS and SARS [18–20]. As such, this study aims to assess the knowledge of, preparedness for, and perceptions toward COVID-19 among healthcare workers in Somalia. These findings will be essential to understand the COVID-19 situation in Somalia, to inform the development of COVID-19 educational programs, and to develop an effective strategy to control the spread of COVID-19 in Somalia.

## Methods

### Design and instrument

Using survey tools from previous published studies and reports assessing knowledge, attitudes, and practices among healthcare workers, a descriptive, cross-sectional survey tool was designed to assess knowledge of and preparedness for COVID-19 among Somali healthcare professionals [21–26]. Our tool assessed healthcare workers' knowledge of COVID-19, access to essential COVID-19 resources, and perception of the Somali government's initial national response to the pandemic. The first section focused on demographic information as well as sources of information on COVID-19 and changes in the number of patients seen since the onset of the pandemic in Somalia. The second section of the survey assessed knowledge of COVID-19 through eight multiple-choice questions regarding COVID-19 transmission, incubation period, symptoms, treatments, and preventative measures. The final section of the survey consisted of 15-items to assess healthcare workers' perceptions of COVID-19, the importance and availability of essential COVID-19 resources, and the strengths and weaknesses in Somalia's national COVID-19 response (**S2 File**).

### Sample and setting

A convenience sample of 364 healthcare workers in Somalia was obtained between June and August 2020. Eligibility requirements for participation in this study included being 18 years or older, physically able to complete the survey, willing to take part in the study, and working within a healthcare profession. Trained staff from the Hagarla Institute, a non-profit organization dedicated to furthering clinical research, capacity-building, and skills transfer for medical personnel across Africa, visited healthcare-delivering institutions within their network in and around Mogadishu and thereafter identified and recruited healthcare professionals who satisfied the aforementioned eligibility requirements. Within each hospital or clinic, Hagarla Institute staff approached potential participants in-person during their working hours to assess eligibility and willingness to participate. If eligibility criteria were met, a brief presentation of the purpose, procedure, and requirements for participation were given privately. Participants were told that they could withdraw from the study at any time and that their participation was entirely voluntary. Given the setting, population of interest, eligibility requirements, and the limited risks associated with participation, verbal, as opposed to written, consent was obtained from participants to facilitate timely recruitment. Verbal, informed consent was witnessed and documented for each participant by Hagarla Institute staff. After receiving consent from participants, Hagarla Institute staff interview-administered each survey. All responses were anonymous and kept confidential. This study received approval from the ethics board at SIMAD University in Somalia and was deemed exempt from review by the Yale Human Subjects Committee (ID #2000028344).

### Data analysis

Responses were analyzed using SAS Studio 3.8. Sample descriptive statistics were used to report survey results, and bivariate and multivariate analyses were carried out to demonstrate the relationships between scores on COVID-19-related questions, gender, age, healthcare profession, having received information on how to best treat COVID-19 patients, number of COVID-19 symptoms correctly identified, and having reported an increase in patient caseload. These findings are presented with 95% confidence intervals. An alpha of 0.05 was adopted for all analyses.

## Results

### Survey demographics

The demographic characteristics of survey participants (n = 364) are shown in Table 1. The majority (60.2%, n = 219) of respondents identified as male. Most participants were comparatively young, with 73.1% (n = 266) under the age of 35 and only 6.0% (n = 22) over the age of 64. Physicians (30.2%, n = 110) and nurses (19.8%, n = 72) were the most well-represented healthcare workers among respondents. Conversely, community health workers only made up 12.1% (n = 44) of the sample.

### Sources of and exposure to COVID-19 information

Respondents' exposure to COVID-19-related information and the specific sources accessed are shown in Table 2. A majority of respondents (69.8%) indicated that they had heard about COVID-19 prior to the confirmation of the first COVID-19 case in Somalia. Similarly, 77.5% of respondents reported that they had received lectures or participated in discussions about COVID-19. Respondents' most accessed forms of information pertaining to COVID-19 were social media (64.8%, n = 236), official government and international health organization websites (51.1%, n = 186), and traditional sources of media such as radio, TV, and newspapers (48.1%, n = 175). Only 18.4% and 19.2% of respondents received information about COVID-19 through academic journals (n = 67) and informational calls or text messages (n = 70), respectively.

### Respondent knowledge of COVID-19

Responses to a series of questions probing participants' knowledge surrounding COVID-19 are shown in Table 3. A majority of participants demonstrated knowledge of the current

**Table 1. Respondent characteristics.**

| Characteristic | N (%)[a,b,] |
|---|---|
| **Sex** | |
| Female | 145 (39.8) |
| Male | 219 (60.2) |
| **Age** | |
| 18–24 | 136 (37.4) |
| 25–34 | 130 (35.7) |
| 35–44 | 38 (10.4) |
| 45–64 | 38 (10.4) |
| 65+ | 22 (6.0) |
| **Profession** | |
| Physician | 110 (30.2) |
| Nurse | 72 (19.8) |
| Midwife | 28 (7.7) |
| Pharmacist | 38 (10.4) |
| Dentist | 15 (4.1) |
| Community Health Worker | 44 (12.1) |
| Other Health Workers[c] | 57 (15.7) |

[a] n = 364.

[b] Values reflect frequency (percentage).

[c] Other health workers included medical students, public health professionals, laboratory workers, etc. One response, from a self-identified shopkeeper, was excluded.

**Table 2. Exposure to COVID-19 information and specific sources accessed among Somali healthcare workers.**

| Source | N (%)[a,b] |
|---|---|
| **Exposure to COVID-19 Information** | |
| Have heard about COVID-19 | 364 (100) |
| Heard about COVID-19 before March 26th (date of first confirmed case in Somalia) | 254 (69.8) |
| Have received lectures or discussions about COVID-19 | 282 (77.5) |
| **Sources of Information** | |
| News, Media (e.g. TV, Radio, Newspapers) | 175 (48.1) |
| Informational calls/SMS | 70 (19.2) |
| Social Media (e.g. Facebook, Twitter, WhatsApp, YouTube, Instagram, Snapchat) | 236 (64.8) |
| Official Government/International Websites (e.g. MoH, DoH, WHO, CDC) | 186 (51.1) |
| Family Members, Colleagues, Friends | 67 (18.4) |
| Employer, Work Colleagues, and Others at Work | 55 (15.1) |
| Non-Governmental Organizations (NGOs) | 58 (15.9) |
| Local or Community Leaders | 60 (16.5) |
| Academic Journals | 67 (18.4) |
| Other Sources of Information | 9 (2.5) |

[a] n = 364.

[b] Values reflect frequency (percentage).

possible treatment of COVID-19, the severity of COVID-19, and the possible outcomes of COVID-19. However, less than a third of participants correctly identified fecal-oral transmission as a way of spreading COVID-19 (26.4%, n = 96). At least 77% of participants correctly identified the main methods of preventing COVID-19 transmission, but only 66.2% of participants identified death as a possible outcome for patients infected with COVID-19 (n = 241). While 8 out of 10 of the symptoms of COVID-19 listed above were identified by a majority of respondents, only "Shortness of Breath or Difficulty Breathing," "Dry Cough," and "Fever" were correctly identified by over 80% of participants. In particular, "Diarrhea" (30.2%, n = 110) and "Runny Nose or Nasal Congestion" (36.5%, n = 133) were the least frequently identified COVID-19 symptoms.

## Treatment of COVID-19 and accessibility of medical resources

The frequency of COVID-19 patients seen by respondents and respondents' access to preventative and treatment resources is shown in Table 4. Notably, 62.1% (n = 226) of respondents reported seeing more patients since March 26th and saw a median of 10 patients with COVID-19 symptoms per week, although this number varied substantially. The most common symptoms seen in suspected COVID-19 patients were shortness of breath (84.6%, n = 308), fever (82.4%, n = 300), dry cough (77.2%, n = 281), and new loss of taste and/or smell (75.6%, n = 275) (S1 File). Despite the volume of patients suspected to have COVID-19, respondents generally reported a scarcity of medical resources required to safely treat patients. In particular, only 21.4% and 30.2% of respondents had access to disposable gowns (n = 78) and N95 masks (n = 110), respectively. The three resources identified by participants as the most important to protect oneself against exposure to COVID-19 were also reported by participants to be the most easily accessible resources, with 65.1%, 58.2%, and 58.2% reporting that it would be somewhat easy or very easy to acquire hand sanitizer (n = 237), disposable gloves (n = 212), and disposable masks (n = 212).

**Table 3. Knowledge of COVID-19 among Somali healthcare workers.**

| Question[b] | Correct N (%)[a,c] | Incorrect/Don't Know N (%)[a,c] |
|---|---|---|
| **Is COVID-19 transmitted through:** | | |
| Airborne Transmission (Yes) | 348 (95.6) | 16 (4.4) |
| Direct Contact of Bodily Fluids (Yes) | 235 (64.6) | 129 (35.4) |
| Waterborne Transmission (No) | 320 (87.9) | 44 (12.1) |
| Fecal-Oral Transmission (Yes) | 96 (26.4) | 268 (73.6) |
| **What is the incubation period of COVID-19?** | | |
| 2–14 days (Yes) | 146 (40.1) | 218 (59.9) |
| **How would you evaluate the severity of COVID-19?** | | |
| Severe Disease which can be fatal in certain cases (Yes) | 260 (71.4) | 104 (28.6) |
| **What is the current possible treatment of COVID-19?** | | |
| Supportive Care (Yes) | 317 (87.1) | 47 (12.9) |
| **Can COVID-19 lead to:** | | |
| Pneumonia (Yes) | 252 (69.2) | 112 (30.8) |
| Respiratory Failure (Yes) | 329 (90.4) | 35 (9.6) |
| Death (Yes) | 241 (66.2) | 123 (33.8) |
| **Can one reduce the risk of COVID-19 transmission by:** | | |
| Hand hygiene (Yes) | 354 (97.3) | 10 (2.7) |
| Covering their nose & face when coughing (Yes) | 295 (81.0) | 69 (19.0) |
| Freezing food that may be contaminated (No) | 303 (83.2) | 61 (16.8) |
| Avoiding places where a large number of people are gathering (Yes) | 294 (80.8) | 70 (19.2) |
| Avoiding sick contacts (Yes) | 283 (77.8) | 81 (22.3) |
| **Do symptoms of COVID-19 include:** | | |
| Headache (Yes) | 275 (75.6) | 89 (24.4) |
| Fever (Yes) | 334 (91.8) | 30 (8.2) |
| Dry Cough (Yes) | 339 (93.1) | 25 (6.9) |
| Sore Throat (Yes) | 245 (67.3) | 119 (32.7) |
| Runny Nose or Nasal Congestions (Yes) | 133 (36.5) | 231 (63.5) |
| New Loss of Taste and/or Smell (Yes) | 281 (77.2) | 83 (22.8) |
| Shortness of Breath or Difficulty Breathing (Yes) | 318 (87.4) | 46 (12.6) |
| Diarrhea (Yes) | 110 (30.2) | 254 (69.8) |
| Muscle or Body Aches (Yes) | 195 (53.6) | 169 (46.4) |
| Fatigue or Malaise (Yes) | 185 (50.8) | 179 (49.2) |

[a] n = 364.

[b] Correct answers displayed.

[c] Values reflect frequency (percentage).

## Evaluation of national action against COVID-19

Respondents' level of agreement with statements evaluating Somali government's initial response to the COVID-19 pandemic are presented in Table 5. While 71.2% of respondents agreed that the Somali lockdown was effective in reducing the spread of COVID-19 (n = 259), 31.3% agreed that Somalia was in a good position to contain an emerging outbreak of COVID-19 (n = 114). Moreover, despite support for a lockdown, participants expressed that the lockdown did not make treatment services more accessible, with only 36.8% (n = 134) and 28.8% (n = 105) reporting that the lockdown made it easier for COVID-19 patients and non-

**Table 4. Patient frequency and access to medical resources among Somali healthcare workers.**

| Source | N (%)[a,b] |
|---|---|
| **Patient Frequency** | |
| Number who reported an increase in patients seen since March 26th | 226 (62.1) |
| Patients seen with COVID-19 symptoms per week | 10 ± 128 |
| **Access to Medical Resources** | 237 (65.1) |
| Hand Sanitizer | 212 (58.2) |
| Disposable Gloves | 78 (21.4) |
| Disposable Gowns | 212 (58.2) |
| Disposable Masks | 110 (30.2) |
| N95 Masks | 89 (24.5) |
| Facial Protective Shields | 103 (28.3) |
| Telemedicine Capacities | 103 (28.3) |

[a] n = 364.

[b] Values reflect frequency (percentage) for categorical variables and median ± standard deviation for continuous variables.

COVID-19 patients with pre-existing conditions to receive treatment, respectively. Furthermore, less than half of respondents agreed that the Somali government's response to the pandemic was sufficient to protect Somalia's residents and healthcare professionals, with only 45.3% (n = 165) and 40.4% (n = 114) of participants affirming these statements, respectively.

## Bi- and multivariate analyses examining high scores on COVID-19 knowledge-related questions

In an adjusted model shown in Table 6, respondents who were fairly young, who did not see more patients since the first confirmed Somali case of COVID-19, and who correctly identified more COVID-19 symptoms displayed significantly higher odds of scoring at or above 75% on

**Table 5. Evaluation of national action against COVID-19 among Somali healthcare workers.**

| Statement | Agree or Strongly Agree N(%)[a,b] | Disagree or Strongly Disagree N (%)[a,b] | Not Sure N (%)[a,b] |
|---|---|---|---|
| The Somali lockdown has been effective in reducing cases and transmission of COVID-19 | 259 (71.2) | 89 (24.5) | 16 (4.4) |
| The Somali lockdown has made it easier for COVID-19 patients to receive continued treatment | 134 (36.8) | 151 (41.5) | 79 (21.7) |
| The Somali lockdown has made it easier for patients without COVID-19 and with pre-existing conditions to receive continued treatment | 105 (28.8) | 177 (48.6) | 82 (22.5) |
| I am aware of Somalia's public health response to the COVID-19 outbreak | 284 (78.0) | 51 (14.0) | 29 (8.0) |
| The Somali government is doing enough to protect its residents from an emerging COVID-19 outbreak | 165 (45.3) | 132 (36.3) | 67 (18.4) |
| The Somali government is doing enough to protect its healthcare professionals from an emerging COVID-19 outbreak | 147 (40.4) | 145 (39.8) | 72 (19.8) |
| Somalia is in a good position to contain an emerging COVID-19 outbreak | 114 (31.3) | 144 (39.6) | 106 (29.1) |

[a] n = 364.

[b] Values reflect frequency (percentage).

**Table 6. Bivariate and multivariable associations between study variables and scores of 75% or higher on a set of questions probing knowledge of COVID-19 among Somali healthcare workers.**

| Characteristic | N (364) | N(%) scoring at or over 75% on a set of questions probing knowledge of COVID-19 | Unadjusted OR (95% CI) | Un- adjusted p-value | Adjusted OR (95% CI) | Adjusted p-value |
|---|---|---|---|---|---|---|
| **Gender** | | | | | | |
| Male | 219 | 111 (50.7) | 1.00 | | 1.00 | |
| Female | 145 | 83 (57.2) | 1.30 (0.85–1.99) | 0.220 | - | - |
| **Age (years)** | | | | | | |
| 35+ | 98 | 66 (67.4) | 1.00 | | 1.00 | |
| 18–34 | 266 | 128 (48.1) | 2.22 (1.37–3.61) | 0.001 | 1.94 (1.14–3.31) | 0.015 |
| **Profession** | | | | | | |
| Other | 182 | 91 (50.0) | 1.00 | | 1.00 | |
| Nurse | 72 | 31 (43.1) | 0.76 (0.44–1.31) | 0.319 | 0.78 (0.42–1.44) | 0.419 |
| Physician | 110 | 72 (65.5) | 1.90 (1.16–3.09) | 0.010 | 1.40 (0.82–2.41) | 0.223 |
| **Received information about how to best handle COVID-19 patients** | | | | | | |
| No | 64 | 36 (56.3) | 1.00 | | 1.00 | |
| Yes | 300 | 158 (52.7) | 0.87 (0.50–1.49) | 0.602 | - | - |
| **Number of COVID-19 symptoms identified** | | | | | | |
| 0–5 | 122 | 30 (24.6) | 1.00 | | 1.00 | |
| 6–7 | 109 | 68 (62.4) | 5.09 (2.89–8.96) | <0.001 | 4.51 (2.52–8.05) | <0.001 |
| 8–10 | 133 | 96 (72.2) | 7.96 (4.55–13.93) | <0.001 | 7.43 (4.17–13.23) | <0.001 |
| **Reporting increased patient volume since first confirmed Somalian case of COVID-19** | | | | | | |
| No | 138 | 85 (61.6) | 1.00 | | 1.00 | |
| Yes | 226 | 109 (48.2) | 0.58 (0.38–0.89) | 0.014 | 0.61 (0.38–1.00) | 0.048 |

questions assessing knowledge of COVID-19 (Table 3). Specifically, survey participants who were 34 years of age or younger had almost twice the odds to have scored 75% or higher (1.94; 95% CI: 1.14–3.31). Interestingly, respondents who reported seeing more patients since the diagnosis of the first COVID-19 patient in Somalia had just over half the odds of scoring at or over 75% on a set of COVID-19 related questions compared to respondents who did not report seeing more patients (0.61; 95% CI: 0.38–1.00). Lastly, compared to those who correctly identified only 0–5 symptoms of COVID-19, respondents who identified 6–7 symptoms of COVID-19 had over four times the odds to have correctly answered at least 75% of COVID-19 knowledge-related questions (4.51; 95% CI: 2.52–8.05), and respondents who identified 8–10 symptoms of COVID-19 had over seven times higher odds to have scored at or higher than 75% (7.43; 95% CI: 4.17–13.23).

## Discussion

To our knowledge, this is the first study examining knowledge of COVID-19, access to essential COVID-19 prevention and treatment resources, and attitudes toward Somalia's pandemic

response among Somali healthcare professionals. Somalia is particularly vulnerable to the spread of pandemic disease, with weak public health infrastructure, 2.6 million internally displaced persons [27], and a hampered national response given a political crisis that resulted in the ousting of Prime Minister Hassan Ali Khaire in July of 2020 [28].

Healthcare workers serve at the forefront against COVID-19 and are vital in the fight against its rapid spread. Therefore, it is imperative that all patient-facing healthcare workers be as familiar as possible with modes of transmission, disease progression, established prevention procedures, and treatment strategies [29]. Even while controlling for inadequate personal protective equipment, healthcare workers remain at an increased risk of becoming infected [30]. While some large studies in the United States have estimated that almost 40% of healthcare workers are at an elevated risk for developing severe COVID-19 illness [31], our sample's demographics suggest that the healthcare workforce in Somalia is not facing the same level of risk for developing severe COVID-19 disease [32], as 73.1% of respondents were younger than 35 and only 6.0% were 65 or older. These results should not discount the importance of ensuring proper facilities and personal protective equipment access to all Somali healthcare workers, as other underlying risk factors such as smoking history may increase vulnerability to developing severe COVID-19 disease among young people to almost one in three [33]. While 62.1% of respondents reported having seen an increase of patients since the beginning of the pandemic, the current status of personal protective equipment availability in Somalia remains low; only hand sanitizer, disposable gloves, and disposable masks were reported as widely accessible by a majority of respondents.

Our study demonstrated significant variation in terms of COVID-19-related knowledge. While the vast majority of respondents were able to correctly identify dry cough as a symptom of COVID-19 (93.1%), that the virus is spread through airborne transmissions by way of respiratory droplets (95.6%), and that the virus could lead to respiratory failure (90.4%), less than half of participants correctly identified COVID-19's incubation period (40.1%) or fecal-oral as a route of transmission for the virus (26.4%). In addition, only about two in three respondents believed that COVID-19 could lead to death (66.2%). In our multivariate model, being under 35, correctly identifying 5 or more COVID-19 symptoms, and having not seen an increase in patients since March 2020 were predictive of correctly answering COVID-19 knowledge questions. In comparison to studies examining COVID-19 knowledge among healthcare workers in neighboring countries, results continue to show significant variability. A study of 442 healthcare workers in Ethiopia discovered that only 66% of respondents believed that COVID-19 may be transmitted through respiratory droplets and 52% were able to accurately identify the incubation period of the disease [34]. Meanwhile, another study of 408 healthcare workers in Northwest Ethiopia found that 94.9% were able to correctly identify COVID-19's incubation period [35]. Novel methods to disseminate COVID-19 related information are necessary. This investigation and the study conducted in Northwest Ethiopia both report that social media, traditional news media, and official governmental and non-governmental reports were the top three sources of COVID-19 information for healthcare workers [35]. As such, utilizing social media and traditional outlets such as television and radio may be effective in circulating key COVID-19 information as a part of national information campaigns. These outlets may be used to target groups with less COVID-19 knowledge to ensure that the entire healthcare workforce is properly equipped with the necessary information to combat the COVID-19 pandemic.

In addition to improving the dissemination of information about COVID-19, these data demonstrate that the Somali government should more effectively share their response plan with healthcare workers. Less than 80% of respondents indicated that they were familiar with the Somali government's national response to combat the COVID-19 pandemic in their own

country. In addition, 45.3% of respondents reported that Somalia was not doing enough to protect its residents from the emerging disease, while just over 40% believed that it was doing enough to protect its healthcare professionals. Of the existing preventive measures instituted by the Somali government, there is preliminary evidence of declining adherence, leading to more widespread infections [36]. In addition, qualitative research has shown that, without financial support, many Somali residents are having their livelihoods disparaged by structural and social factors and burdened by lockdown and disease transmission [37]. The Somali government needs to build upon its most recent quest to disseminate information with the launching of the toll-free #449 hotline, which connects over 3,000 callers per day with healthcare professionals to answer COVID-19-related questions [38].

## Limitations

This study is subject to various limitations. First, this study utilized a convenience sample, which is likely not representative of the entire Somali healthcare workforce. In addition, the sampling method's reliance on connections to the Hagarla Institute may produce a sampling bias geared towards healthcare workers located closer to Mogadishu. These biases may be partially responsible for our sample's younger age profile. Similarly, given challenges associated with data collection in the field, especially during a COVID-19 wave, this study did not collect information relating to non-respondents. As such, this may also bias the generalizability of the study findings. Furthermore, because our multivariate analysis did not adjust for potential confounding factors, residual confounding may have been introduced. Finally, as this study utilized a cross-sectional design, it cannot be used to analyze the evolution of COVID-19 knowledge, access to personal protective equipment, and perceptions of the Somali national response over an extended period of time. This study took place during Somalia's first wave of COVID-19. Since then, larger waves have been recorded, which is expected to have impacted knowledge of and preparedness for COVID-19 among Somali healthcare professionals. Although this study does not capture the progression of Somalia's COVID-19 situation, the study findings provide valuable insights into pertinent factors that must be considered in preparing for and choosing how to disseminate information during future outbreaks. These findings also have direct implications for the development of future research priorities and interventions related to COVID-19 in Somalia, as well as other similar resource-constrained settings. This is a context where very little methodologically rigorous research has been conducted, and we are hopeful that this study will serve as a reference point for future public health and research initiatives being developed currently and beyond the COVID-19 pandemic.

## Conclusion

This study provides evidence for the need to improve access to COVID-19 information among healthcare professionals in Somalia, a nation that is ill-equipped to resolve dramatic health inequities due to inadequate health infrastructure and shortages of healthcare personnel. It highlights a perception of inadequate access to telemedicine services and basic personal protective equipment, such as disposable gowns and N95 masks, in Somalia. This investigation also reveals a disconnect between healthcare workers and the Somali national response, which was widely perceived as failing to protect residents and healthcare workers from COVID-19. In order to sufficiently communicate with its frontline workers about developing COVID-19 information and the government's response, the Somali government must utilize the channels healthcare workers identified as highly utilized, namely social media, traditional news outlets, and the official governmental website.

## Supporting information

**S1 File. Dataset.**
(XLSX)

**S2 File. Study questionnaire in English.**
(DOCX)

## Acknowledgments

We are thankful to the Hagarla Institute and its staff for their support in coordinating this important work.

## Author Contributions

**Conceptualization:** Jude Alawa, Lucas Walz, Samir Al-Ali, Nikhil Harle, Eleanor Wiles, Mohamed Abdullahi Awale, Deqo Mohamed, Kaveh Khoshnood.

**Data curation:** Jude Alawa, Lucas Walz, Samir Al-Ali, Nikhil Harle, Eleanor Wiles, Mohamed Abdullahi Awale, Deqo Mohamed, Kaveh Khoshnood.

**Formal analysis:** Jude Alawa, Lucas Walz, Samir Al-Ali, Nikhil Harle, Kaveh Khoshnood.

**Funding acquisition:** Jude Alawa, Lucas Walz, Samir Al-Ali, Nikhil Harle, Mohamed Abdullahi Awale, Deqo Mohamed, Kaveh Khoshnood.

**Investigation:** Jude Alawa, Lucas Walz, Samir Al-Ali, Nikhil Harle, Eleanor Wiles, Mohamed Abdullahi Awale, Deqo Mohamed, Kaveh Khoshnood.

**Methodology:** Jude Alawa, Lucas Walz, Samir Al-Ali, Eleanor Wiles, Mohamed Abdullahi Awale, Deqo Mohamed, Kaveh Khoshnood.

**Project administration:** Jude Alawa, Eleanor Wiles, Mohamed Abdullahi Awale, Deqo Mohamed.

**Resources:** Jude Alawa, Mohamed Abdullahi Awale, Deqo Mohamed, Kaveh Khoshnood.

**Software:** Lucas Walz, Nikhil Harle.

**Supervision:** Mohamed Abdullahi Awale, Deqo Mohamed, Kaveh Khoshnood.

**Validation:** Jude Alawa, Lucas Walz, Samir Al-Ali, Nikhil Harle, Eleanor Wiles, Mohamed Abdullahi Awale, Deqo Mohamed, Kaveh Khoshnood.

**Visualization:** Jude Alawa, Lucas Walz, Nikhil Harle, Mohamed Abdullahi Awale, Kaveh Khoshnood.

**Writing – original draft:** Jude Alawa, Lucas Walz, Samir Al-Ali, Nikhil Harle, Eleanor Wiles, Deqo Mohamed.

**Writing – review & editing:** Jude Alawa, Lucas Walz, Samir Al-Ali, Nikhil Harle, Eleanor Wiles, Mohamed Abdullahi Awale, Deqo Mohamed, Kaveh Khoshnood.

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
