## [Decision Letter · Decision Letter 0]

30 Jun 2021

PONE-D-21-11430

Knowledge of and preparedness for COVID-19 among Somali healthcare professionals: a cross-sectional study

PLOS ONE

Dear Dr. Khoshnood,

Thank you for submitting your manuscript to PLOS ONE. After careful consideration, we feel that it has merit but does not fully meet PLOS ONE’s publication criteria as it currently stands. Therefore, we invite you to submit a revised version of the manuscript that addresses the points raised during the review process.

We look forward to receiving your revised manuscript.

Kind regards,

Jianguo Wang, PhD

Academic Editor

PLOS ONE

Additional Editor Comments:

The recommendations from two reviewers are mixed. Please confirm the relability of data and also pay attentions to the individual privacy.

Journal Requirements:

2. Please provide additional details regarding participant consent. In the ethics statement in the Methods and online submission information, please describe how verbal consent was documented and witnessed, and why written consent was not obtained. Please also state whether consent was informed.

3. Thank you for sending us the data set underlying the results presented in your PLOS ONE submission. We notice that some of the information included in the supplemental data set may be potentially identifying. Please ensure that the data shared are in accordance with participant consent and provide only the data that are used in this specific study. To ensure patient confidentiality, we would recommend removing the columns with IP address, latitude, and longitude. Additional guidance on preparing raw clinical data for publication can be found in our Data Policy FAQs (https://journals.plos.org/plosone/s/data-availability#loc-clinical-data).

4. Please include additional information regarding the survey or questionnaire used in the study and ensure that you have provided sufficient details that others could replicate the analyses. For instance, if you developed the survey or questionnaire as part of this study and it is not under a copyright more restrictive than CC-BY, please include a copy, in both the original language and English, as Supporting Information. If the questionnaire is published, please provide a citation to the (1) questionnaire and/or (2) original publication associated with the questionnaire.

Reviewers' comments:

Reviewer's Responses to Questions

**Comments to the Author**

1. Is the manuscript technically sound, and do the data support the conclusions?

Reviewer #1: Yes

Reviewer #2: Partly

2. Has the statistical analysis been performed appropriately and rigorously? 

Reviewer #1: I Don't Know

Reviewer #2: Yes

3. Have the authors made all data underlying the findings in their manuscript fully available?

Reviewer #1: Yes

Reviewer #2: Yes

4. Is the manuscript presented in an intelligible fashion and written in standard English?

Reviewer #1: Yes

Reviewer #2: Yes

5. Review Comments to the Author

Reviewer #1: This is a nicely conceived, executed and written up research project that is worthy of publication due to its sound and interesting findings. It should be tweaked presentationally in some small respects. It says "Moreover, only 31.3% agreed that Somalia was in a good position to contain an emerging outbreak of COVID-19. In addition, only 40.4% of participants agreed that the Somali government’s response to the pandemic was sufficient to protect Somali healthcare professionals." Considering Somalia's extreme underpreparedness, it is a puzzle that so many participants agreed that the Somalia was in a good position to contain the outbreak or protect Somali healthcare professionals and so in both cases 'only' seems to be the wrong word. My speculation is that the participants had such low expectations that they were impressed by what little was done.

Reviewer #2: It is a bit surprising that this questionnaire was completed in 2020 and the results are only now sent out for publication. According to (https://www.worldometers.info/coronavirus/country/somalia/) Somalia was experiencing its first COVID wave in June 2020, albeit with a low number of cases reported per day. However, since then a larger wave was recorded and this would be expected to have impacted the knowledge and perceptions of the healthcare workers, making the information collected through this survey somewhat outdated.

In order to gather data that would be more relevant to the current moment in the pandemic’s course, I would encourage the authors to reapply the questionnaire in summer 2021 and to compare the results with those of the survey from 2020.

Furthermore, it would be important to also present data on how the questionnaire was applied, who was invited to participate, how the invitations were sent and what fraction of the invitations were accepted. Also, if there is information regarding non-respondents, it would important to also present this data, in order to better understand to whom this information is generalizable.

As a side note, I am a bit worried about the authors choosing to share as supplementary file the excel file which includes IP address as well as longitude and latitude, as this may lead to identification of participants, or their home address.

6. PLOS authors have the option to publish the peer review history of their article (what does this mean?). If published, this will include your full peer review and any attached files.

Reviewer #1: **Yes: **Eric Herring

Reviewer #2: No

---

## [Author Response · Author response to Decision Letter 0]

6 Aug 2021

August 6th, 2021

Dear Editor,

Thank you very much for your review of our manuscript for PLOS One entitled, "Knowledge of and preparedness for COVID-19 among Somali healthcare professionals: a cross-sectional study." We appreciate the editors’ and reviewers’ time and believe the edits have strengthened the manuscript considerably. We have addressed the editors’ and reviewers’ feedback below and have submitted a revised manuscript with tracked changes that correspond to the edits outlined in this letter.

I hope that we have addressed all the concerns and that the manuscript is now suitable for publication. Please do not hesitate to contact me if you have any questions or comments.

Sincerely,

Kaveh Khoshnood, PhD, MPH

Response to Reviewers for “Knowledge of and preparedness for COVID-19 among Somali healthcare professionals: a cross-sectional study”

Comments from the Editor:

Comment 1: The recommendations from two reviewers are mixed. Please confirm the reliability of data and also pay attention to the individual privacy.

Response 1: We thank the reviewers and editors for their thorough review of our manuscript. We understand that the recommendations from the two reviewers are mixed, and we hope that our responses and revisions have improved the manuscript such that it is now suitable for publication. Given the timeliness of this research, the limited research on COVID-19 in the Somali context, and Somalia’s severe underpreparedness to address the COVID-19, we believe our study provides valuable findings that can be used to inform future research and local interventions. We have reviewed the manuscript to confirm the reliability of the data. Per Response 4, we have addressed concerns regarding individual privacy. We have revised the dataset file and can now confirm that the data shared are specific to this study, are in accordance with patient consent, and do not contain any personal identifying information. A new version of our Supporting Information dataset file, entitled “S1_File” has been attached.

Comments regarding Journal Requirements:

Comment 2: 1. Please ensure that your manuscript meets PLOS ONE's style requirements, including those for file naming. The PLOS ONE style templates can be found at

Response 2: Thank you for providing these instructions and guidelines. We have revised the manuscript according to PLOS ONE’s style requirements, including those for file naming. 

Comment 3: Please provide additional details regarding participant consent. In the ethics statement in the Methods and online submission information, please describe how verbal consent was documented and witnessed, and why written consent was not obtained. Please also state whether consent was informed. 

Response 3: Thank you for this important comment. We have added further information in the Methods section and online submission portal to reflect the process by which verbal consent was documented and witnessed, and why written consent was not obtained. We have also confirmed that consent was informed. 

This subsection now reads as: 

“A convenience sample of 364 healthcare workers in Somalia was obtained between June and August 2020. Eligibility requirements for participation in this study included being 18 years or older, physically able to complete the survey, and willing to take part in the study. If these criteria were met, a brief presentation of the purpose, procedure, and requirements for participation were given privately. Participants were told that they could withdraw from the study at any time and that their participation was entirely voluntary. Given the setting, population of interest, eligibility requirements, and the limited risks associated with participation, verbal, as opposed to written, consent was obtained from participants to facilitate timely recruitment. Informed consent was witnessed and documented for each participant by trained staff from the Hagarla Institute, who interview-administered surveys and recruited potential participants from healthcare-delivering institutions in and around Mogadishu, Somalia. The Hagarla Institute is a non-profit organization dedicated to furthering clinical research, capacity-building, and skills transfer for medical personnel across Africa. All responses were anonymous and kept confidential. This study received approval from the ethics board at SIMAD University in Somalia and was deemed exempt from review by the Yale Human Subjects Committee (ID #2000028344).”

Comment 4: Thank you for sending us the data set underlying the results presented in your PLOS ONE submission. We notice that some of the information included in the supplemental data set may be potentially identifying. Please ensure that the data shared are in accordance with participant consent and provide only the data that are used in this specific study. To ensure patient confidentiality, we would recommend removing the columns with IP address, latitude, and longitude. Additional guidance on preparing raw clinical data for publication can be found in our Data Policy FAQs (https://journals.plos.org/plosone/s/data-availability#loc-clinical-data).

Response 4: Thank you for bringing this critical information to our attention. We have revised the dataset file and can confirm that the data shared are specific to this study, are in accordance with patient consent, and do not contain any personal identifying information. A new version of our Supporting Information dataset file, entitled “S1_File” has been attached. 

Comment 5: Please include additional information regarding the survey or questionnaire used in the study and ensure that you have provided sufficient details that others could replicate the analyses. For instance, if you developed the survey or questionnaire as part of this study and it is not under a copyright more restrictive than CC-BY, please include a copy, in both the original language and English, as Supporting Information. If the questionnaire is published, please provide a citation to the (1) questionnaire and/or (2) original publication associated with the questionnaire.

Response 5: Thank you for your comment. We have now attached, with appropriate captioning, another Supporting Information file (S2_File) containing the questionnaire used for this study, which was created and administered in English. We have also ensured that the details provided in the manuscript are sufficient for others to replicate our findings and analyses.

Comment 6: We note that you have included the phrase “data not shown” in your manuscript. Unfortunately, this does not meet our data sharing requirements. PLOS does not permit references to inaccessible data. We require that authors provide all relevant data within the paper, Supporting Information files, or in an acceptable, public repository. Please add a citation to support this phrase or upload the data that corresponds with these findings to a stable repository (such as Figshare or Dryad) and provide and URLs, DOIs, or accession numbers that may be used to access these data. Or, if the data are not a core part of the research being presented in your study, we ask that you remove the phrase that refers to these data.

Response 6: Thank you for bringing this to our attention. To ensure all of our referenced data is available and easily accessible to readers per PLOS One guidelines, we have included the most relevant material in the body of the manuscript and included the remainder of our data in a Supporting Information file containing our entire dataset. The file is called “S1_File.” We have now removed the “data not shown” reference and replaced it with an in-text reference to the Supporting Information file, S1_File. 

Comment 7: Please include captions for your Supporting Information files at the end of your manuscript, and update any in-text citations to match accordingly. Please see our Supporting Information guidelines for more information: http://journals.plos.org/plosone/s/supporting-information.

Response 7: Thank you for your comment. We have now added captions for the Supporting Information files, including our dataset and study questionnaire, at the end of the manuscript. We have also updated in-text citations referring to these files. 

Reviewer comments:

Comment 8: 1. Is the manuscript technically sound, and do the data support the conclusions?The manuscript must describe a technically sound piece of scientific research with data that supports the conclusions. Experiments must have been conducted rigorously, with appropriate controls, replication, and sample sizes. The conclusions must be drawn appropriately based on the data presented.

Reviewer #1: Yes

Reviewer #2: Partly

Response 8: We thank the reviewers for their assessment of our manuscript. We have reviewed the manuscript to ensure that the methods are clearly described and that conclusions drawn based on our findings are technically sound and based on rigorous statistical analyses. We have made all of our data available in the manuscript and Supporting Information files to demonstrate that our conclusions are supported by the data collected and analyses described. 

Comment 9: 2. Has the statistical analysis been performed appropriately and rigorously?

Reviewer #1: I Don't Know

Reviewer #2: Yes

Response 9: We thank the reviewers for their thorough review of our analyses. Our statistical analysis utilized conventional bivariate and multivariate models to rigorously test correlations between scores on COVID-19-related questions, gender, age, healthcare profession, having received information on how to best treat COVID-19 patients, number of COVID-19 symptoms correctly identified, and having reported an increase in patient caseload. All of our statistical findings are presented in their entirety, with all variables, measures of correlation, and confidence intervals appropriately and clearly notated. 

Comment 10: 3. Have the authors made all data underlying the findings in their manuscript fully available? The PLOS Data policy requires authors to make all data underlying the findings described in their manuscript fully available without restriction, with rare exception (please refer to the Data Availability Statement in the manuscript PDF file). The data should be provided as part of the manuscript or its supporting information, or deposited to a public repository. For example, in addition to summary statistics, the data points behind means, medians and variance measures should be available. If there are restrictions on publicly sharing data—e.g. participant privacy or use of data from a third party—those must be specified.

Reviewer #1: Yes

Reviewer #2: Yes

Response 10: Thank you for your attention to this. We can confirm as well that all data underlying the finds in our manuscript have been made fully available. 

Comment 11: 4. Is the manuscript presented in an intelligible fashion and written in standard English? PLOS ONE does not copyedit accepted manuscripts, so the language in submitted articles must be clear, correct, and unambiguous. Any typographical or grammatical errors should be corrected at revision, so please note any specific errors here.

Reviewer #1: Yes

Reviewer #2: Yes

Response 11: We thank the reviewers for their thorough review of our manuscript. We have proofread the manuscript once again to confirm that the language used is presented in an intelligible fashion and written in standard English. 

Comment 12: Reviewer #1 

Comment 12.1: This is a nicely conceived, executed and written up research project that is worthy of publication due to its sound and interesting findings. It should be tweaked presentationally in some small respects. It says "Moreover, only 31.3% agreed that Somalia was in a good position to contain an emerging outbreak of COVID-19. In addition, only 40.4% of participants agreed that the Somali government’s response to the pandemic was sufficient to protect Somali healthcare professionals." Considering Somalia's extreme underpreparedness, it is a puzzle that so many participants agreed that the Somalia was in a good position to contain the outbreak or protect Somali healthcare professionals and so in both cases 'only' seems to be the wrong word. My speculation is that the participants had such low expectations that they were impressed by what little was done.

Response 12.1: Thank you for your favorable review of our manuscript and for your comments. We are deeply appreciative of your time, and we are eager to share our findings on this timely and important topic in Somalia, a country that is facing monumental challenges in addressing COVID-19’s impact on their population’s health and safety. With respect to your comment, we have edited the sentences mentioned above to remove the word “only” in each case. 

Comment 13: Reviewer #2: 

Comment 13.1: It is a bit surprising that this questionnaire was completed in 2020 and the results are only now sent out for publication. According to (https://www.worldometers.info/coronavirus/country/somalia/) Somalia was experiencing its first COVID wave in June 2020, albeit with a low number of cases reported per day. However, since then a larger wave was recorded and this would be expected to have impacted the knowledge and perceptions of the healthcare workers, making the information collected through this survey somewhat outdated. In order to gather data that would be more relevant to the current moment in the pandemic’s course, I would encourage the authors to reapply the questionnaire in summer 2021 and to compare the results with those of the survey from 2020.

Response 13.1: Thank you for your review of our manuscript and for your important comments and suggestions. We are deeply appreciative of your time and consideration. We recognize that the data used for this study was collected in 2020 and is now being considered for publication. Several of our research team members have been working hard to actively address the COVID-19 situation on-the-ground in Somalia and to utilize our study findings to inform future research initiatives and interventions in the field. While we recognize that larger waves of COVID-19 have been recorded since our data has been collected, we strongly believe that the information presented in our manuscript is still timely and can provide health fieldworkers with valuable insight into Somali healthcare professionals’ knowledge of and preparedness for COVID-19. In addition, we believe our findings are relevant to understanding the health situation in Somalia beyond solely the COVID-19 pandemic. For example, the findings indicating healthcare professionals’ preferred sources by which to receive developing and novel information may expedite the sharing of information in future outbreaks, COVID-19-related or otherwise. 

Furthermore, our research has direct implications for the development of future research priorities and interventions related to COVID-19 in Somalia, as well as other similar resource-constrained settings. This is a context where very little methodologically rigorous research has been conducted, and we are hopeful that our study will serve as a reference point for future public health and research initiatives being developed currently and beyond the COVID-19 pandemic. We are grateful for the reviewer’s suggestion to reapply the questionnaire as our team has been continuing to work in the field on research efforts to further elucidate the COVID-19 situation in Somalia. 

Comment 13.2: Furthermore, it would be important to also present data on how the questionnaire was applied, who was invited to participate, how the invitations were sent and what fraction of the invitations were accepted. Also, if there is information regarding non-respondents, it would important to also present this data, in order to better understand to whom this information is generalizable.

Response 13.2: Thank you very much for your important suggestions. In the Methods section, under Sample and Setting, we have added further information about how the questionnaire was applied, who was invited to participate, and how participants were recruited. Unfortunately, given challenges associated with data collection in the field, especially during a COVID-19 wave, we do not have data regarding non-respondents, which may bias the generalizability of our findings. We recognize this as a limitation and have incorporated it into the Limitations section within the manuscript. 

Comment 13.3: As a side note, I am a bit worried about the authors choosing to share as supplementary file the excel file which includes IP address as well as longitude and latitude, as this may lead to identification of participants, or their home address.

Response 13.3: We thank the reviewer for bringing our attention to this critical point. In accordance with Response 4, we have revised the dataset file, and we can confirm that the data shared are specific to this study, are in accordance with patient consent, and do not contain any personal identifying information. We have removed any information relating to IP address, latitude, and longitude. A new version of our Supporting Information dataset file, entitled “S1_File” has been attached.

---

## [Decision Letter · Decision Letter 1]

1 Oct 2021

PONE-D-21-11430R1Knowledge of and preparedness for COVID-19 among Somali healthcare professionals: a cross-sectional studyPLOS ONE

Dear Dr. Khoshnood,

Thank you for submitting your manuscript to PLOS ONE. After careful consideration, we feel that it has merit but does not fully meet PLOS ONE’s publication criteria as it currently stands. Therefore, we invite you to submit a revised version of the manuscript that addresses the points raised during the review process.

We look forward to receiving your revised manuscript.

Kind regards,

Jianguo Wang, PhD

Academic Editor

PLOS ONE

Journal Requirements:

Additional Editor Comments (if provided):

Reviewers' comments:

Reviewer's Responses to Questions

**Comments to the Author**

1. If the authors have adequately addressed your comments raised in a previous round of review and you feel that this manuscript is now acceptable for publication, you may indicate that here to bypass the “Comments to the Author” section, enter your conflict of interest statement in the “Confidential to Editor” section, and submit your "Accept" recommendation.

Reviewer #2: (No Response)

2. Is the manuscript technically sound, and do the data support the conclusions?

Reviewer #2: Yes

3. Has the statistical analysis been performed appropriately and rigorously? 

Reviewer #2: Yes

4. Have the authors made all data underlying the findings in their manuscript fully available?

Reviewer #2: Yes

5. Is the manuscript presented in an intelligible fashion and written in standard English?

Reviewer #2: Yes

6. Review Comments to the Author

Reviewer #2: - I would suggest that the authors include their response to my previous query, or at least a part of it, in the Study limitations section, so that it is also available to the readers: Response 13.1.

- Sample and setting: The following information has been added: “If these criteria were met, a brief presentation of the purpose, procedure, and requirements for participation were given privately.” However, it is still not clear to whom this presentation was given? How were these participants identified and chosen? From a specific hospital? Hospital network? Professional society? From social media? A hint on this is given in the lines below, but it is still not clear: “recruited potential participants within healthcare-delivering institutions in and around Mogadishu, Somalia”

7. PLOS authors have the option to publish the peer review history of their article (what does this mean?). If published, this will include your full peer review and any attached files.

Reviewer #2: No

---

## [Author Response · Author response to Decision Letter 1]

6 Oct 2021

October 6th, 2021

Dear Editor,

Thank you very much for your review of our manuscript for PLOS One entitled, "Knowledge of and preparedness for COVID-19 among Somali healthcare professionals: a cross-sectional study." We appreciate the editors’ and reviewers’ time and believe the edits have strengthened the manuscript considerably. We have addressed the editors’ and reviewers’ feedback below and have submitted a revised manuscript with tracked changes that correspond to the edits outlined in this letter.

I hope that we have addressed all the concerns and that the manuscript is now suitable for publication. Please do not hesitate to contact me if you have any questions or comments.

Sincerely,

Kaveh Khoshnood, PhD, MPH

Response to Reviewers for “Knowledge of and preparedness for COVID-19 among Somali healthcare professionals: a cross-sectional study”

Reviewer comments:

Comment 1: I would suggest that the authors include their response to my previous query, or at least a part of it, in the Study limitations section, so that it is also available to the readers: Response 13.1.

Response 1: Thank you for this important suggestion. We have incorporated our response (Response 13.1) to your previous query in our Limitations section. 

The section now reads: 

"This study is subject to various limitations. First, this study utilized a convenience sample, which is likely not representative of the entire Somali healthcare workforce. In addition, the sampling method’s reliance on connections to the Hagarla Institute may produce a sampling bias geared towards healthcare workers located closer to Mogadishu. These biases may be partially responsible for our sample’s younger age profile. Similarly, given challenges associated with data collection in the field, especially during a COVID-19 wave, this study did not collect information relating to non-respondents. As such, this may also bias the generalizability of the study findings. Because our multivariate analysis did not adjust for potential confounding factors, residual confounding may also have been introduced. Furthermore, as this study utilized a cross-sectional design, it cannot be used to analyze the evolution of COVID-19 knowledge, access to personal protective equipment, and perceptions of the Somali national response over an extended period of time. This study took place during Somalia’s first wave of COVID-19. Since then, larger waves have been recorded, which is expected to have impacted knowledge of and preparedness for COVID-19 among Somali healthcare professionals. Although this study does not capture the progression of Somalia’s COVID-19 situation, the study findings provide valuable insights into pertinent factors that must be considered in preparing for and choosing how to disseminate information during future outbreaks. These findings also have direct implications for the development of future research priorities and interventions related to COVID-19 in Somalia, as well as other similar resource-constrained settings. This is a context where very little methodologically rigorous research has been conducted, and we are hopeful that this study will serve as a reference point for future public health and research initiatives being developed currently and beyond the COVID-19 pandemic."

Comment 2: Sample and setting: The following information has been added: “If these criteria were met, a brief presentation of the purpose, procedure, and requirements for participation were given privately.” However, it is still not clear to whom this presentation was given? How were these participants identified and chosen? From a specific hospital? Hospital network? Professional society? From social media? A hint on this is given in the lines below, but it is still not clear: “recruited potential participants within healthcare-delivering institutions in and around Mogadishu, Somalia.”

Response 2: Thank you for bringing this point to our attention. We recognize the need for further detail as to how participants were identified and have added further information in the Sample and Setting subsection of the Methods section. 

The subsection now reads: 

"A convenience sample of 364 healthcare workers in Somalia was obtained between June and August 2020. Eligibility requirements for participation in this study included being 18 years or older, physically able to complete the survey, willing to take part in the study, and working within a healthcare profession. If these criteria were met, a brief presentation of the purpose, procedure, and requirements for participation were given privately. Participants were told that they could withdraw from the study at any time and that their participation was entirely voluntary. Given the setting, population of interest, eligibility requirements, and the limited risks associated with participation, verbal, as opposed to written, consent was obtained from participants to facilitate timely recruitment. Verbal, informed consent was witnessed and documented for each participant by trained staff from the Hagarla Institute, a non-profit organization dedicated to furthering clinical research, capacity-building, and skills transfer for medical personnel across Africa. Hagarla Institute personnel visited healthcare-delivering institutions within their network in and around Mogadishu and identified and recruited healthcare professionals who satisfied the aforementioned eligibility requirements. After receiving consent from participants, Hagarla Institute staff interview-administered each survey. All responses were anonymous and kept confidential. This study received approval from the ethics board at SIMAD University in Somalia and was deemed exempt from review by the Yale Human Subjects Committee (ID #2000028344)."

---

## [Decision Letter · Decision Letter 2]

12 Oct 2021

PONE-D-21-11430R2Knowledge of and preparedness for COVID-19 among Somali healthcare professionals: a cross-sectional studyPLOS ONE

Dear Dr. Khoshnood,

Thank you for submitting your manuscript to PLOS ONE. After careful consideration, we feel that it has merit but does not fully meet PLOS ONE’s publication criteria as it currently stands. Therefore, we invite you to submit a revised version of the manuscript that addresses the points raised during the review process.

ACADEMIC EDITOR: Please insert comments here and delete this placeholder text when finished. Be sure to:Please carefully address teh comments from reviewers.==============================

We look forward to receiving your revised manuscript.

Kind regards,

Jianguo Wang, PhD

Academic Editor

PLOS ONE

Journal Requirements:

Reviewers' comments:

Reviewer's Responses to Questions

**Comments to the Author**

1. If the authors have adequately addressed your comments raised in a previous round of review and you feel that this manuscript is now acceptable for publication, you may indicate that here to bypass the “Comments to the Author” section, enter your conflict of interest statement in the “Confidential to Editor” section, and submit your "Accept" recommendation.

Reviewer #2: (No Response)

2. Is the manuscript technically sound, and do the data support the conclusions?

Reviewer #2: Yes

3. Has the statistical analysis been performed appropriately and rigorously? 

Reviewer #2: Yes

4. Have the authors made all data underlying the findings in their manuscript fully available?

Reviewer #2: Yes

5. Is the manuscript presented in an intelligible fashion and written in standard English?

Reviewer #2: Yes

6. Review Comments to the Author

Reviewer #2: Unfortunately, the authors have not addressed my previous comment:

“Sample and setting: The following information has been added: “If these criteria were met, a brief presentation of the purpose, procedure, and requirements for participation were given privately.” However, it is still not clear to whom this presentation was given? How were these participants identified and chosen? From a specific hospital? Hospital network? Professional society? From social media? A hint on this is given in the lines below, but it is still not clear: “recruited potential participants within healthcare-delivering institutions in and around Mogadishu, Somalia””

I would suggest that the authors briefly explain how they identified potential participants. For example, did they use an existing database of medical personnel from a certain hospital or a certain profession society? Or were participants approached in their own hospital during their working hours? Also, this section should clarify through which means they reached out to study participants, by phone, email, in person, etc.

7. PLOS authors have the option to publish the peer review history of their article (what does this mean?). If published, this will include your full peer review and any attached files.

Reviewer #2: No

---

## [Author Response · Author response to Decision Letter 2]

15 Oct 2021

October 15th, 2021

Dear Editor,

Thank you very much for your review of our manuscript for PLOS One entitled, "Knowledge of and preparedness for COVID-19 among Somali healthcare professionals: a cross-sectional study." We appreciate the editors’ and reviewers’ time and believe the edits have strengthened the manuscript considerably. We have addressed the editors’ and reviewers’ feedback below and have submitted a revised manuscript with tracked changes that correspond to the edits outlined in this letter.

I hope that we have addressed all the concerns and that the manuscript is now suitable for publication. Please do not hesitate to contact me if you have any questions or comments.

Sincerely,

Kaveh Khoshnood, PhD, MPH

Response to Reviewers for “Knowledge of and preparedness for COVID-19 among Somali healthcare professionals: a cross-sectional study”

Reviewer comments:

Comment 1: Unfortunately, the authors have not addressed my previous comment:

“Sample and setting: The following information has been added: “If these criteria were met, a brief presentation of the purpose, procedure, and requirements for participation were given privately.” However, it is still not clear to whom this presentation was given? How were these participants identified and chosen? From a specific hospital? Hospital network? Professional society? From social media? A hint on this is given in the lines below, but it is still not clear: “recruited potential participants within healthcare-delivering institutions in and around Mogadishu, Somalia””

I would suggest that the authors briefly explain how they identified potential participants. For example, did they use an existing database of medical personnel from a certain hospital or a certain profession society? Or were participants approached in their own hospital during their working hours? Also, this section should clarify through which means they reached out to study participants, by phone, email, in person, etc.

Response 1: Thank you very much for this important suggestion. We recognize the need for further detail as to how potential participants were identified. As such, we had added more information to clarify that participants were reached out to in-person and that participants were approached in their own hospital during their working hours. Trained staff from the Hagarla Institute visited health facilities within their network and approached potential participants in-person during their working hours to assess eligibility and willingness to participate. 

The subsection now reads: 

 “A convenience sample of 364 healthcare workers in Somalia was obtained between June and August 2020. Eligibility requirements for participation in this study included being 18 years or older, physically able to complete the survey, willing to take part in the study, and working within a healthcare profession. Trained staff from the Hagarla Institute, a non-profit organization dedicated to furthering clinical research, capacity-building, and skills transfer for medical personnel across Africa, visited healthcare-delivering institutions within their network in and around Mogadishu and thereafter identified and recruited healthcare professionals who satisfied the aforementioned eligibility requirements. Within each hospital or clinic, Hagarla Institute staff approached potential participants in-person during their working hours to assess eligibility and willingness to participate. If eligibility criteria were met, a brief presentation of the purpose, procedure, and requirements for participation were given privately. Participants were told that they could withdraw from the study at any time and that their participation was entirely voluntary. Given the setting, population of interest, eligibility requirements, and the limited risks associated with participation, verbal, as opposed to written, consent was obtained from participants to facilitate timely recruitment. Verbal, informed consent was witnessed and documented for each participant by Hagarla Institute staff. After receiving consent from participants, Hagarla Institute staff interview-administered each survey. All responses were anonymous and kept confidential. This study received approval from the ethics board at SIMAD University in Somalia and was deemed exempt from review by the Yale Human Subjects Committee (ID #2000028344).”

---

## [Decision Letter · Decision Letter 3]

2 Nov 2021

Knowledge of and preparedness for COVID-19 among Somali healthcare professionals: a cross-sectional study

PONE-D-21-11430R3

Dear Dr. Khoshnood,

We’re pleased to inform you that your manuscript has been judged scientifically suitable for publication and will be formally accepted for publication once it meets all outstanding technical requirements.

Kind regards,

Jianguo Wang, PhD

Academic Editor

PLOS ONE

Additional Editor Comments (optional):

Reviewers' comments:

Reviewer's Responses to Questions

**Comments to the Author**

1. If the authors have adequately addressed your comments raised in a previous round of review and you feel that this manuscript is now acceptable for publication, you may indicate that here to bypass the “Comments to the Author” section, enter your conflict of interest statement in the “Confidential to Editor” section, and submit your "Accept" recommendation.

Reviewer #2: All comments have been addressed

2. Is the manuscript technically sound, and do the data support the conclusions?

Reviewer #2: Yes

3. Has the statistical analysis been performed appropriately and rigorously? 

Reviewer #2: Yes

4. Have the authors made all data underlying the findings in their manuscript fully available?

Reviewer #2: Yes

5. Is the manuscript presented in an intelligible fashion and written in standard English?

Reviewer #2: Yes

6. Review Comments to the Author

Reviewer #2: I thank the authors for addressing my previous comments. I now have no further comments at this point.

7. PLOS authors have the option to publish the peer review history of their article (what does this mean?). If published, this will include your full peer review and any attached files.

Reviewer #2: No

---

## [Editor Report · Acceptance letter]

15 Nov 2021

PONE-D-21-11430R3 

Knowledge of and preparedness for COVID-19 among Somali healthcare professionals: a cross-sectional study 

Dear Dr. Khoshnood:

I'm pleased to inform you that your manuscript has been deemed suitable for publication in PLOS ONE. Congratulations! Your manuscript is now with our production department. 

Kind regards, 

on behalf of

Dr. Jianguo Wang 

Academic Editor

PLOS ONE